# Impact of urbanization on thermal environment of Chengdu-Chongqing Urban Agglomeration under complex terrain

Si Chen[1,2], Zhenghui Xie[1,2*], Jinbo Xie[1], Bin Liu[3], Binghao Jia[1], Peihua Qin[1], Longhuan Wang[1,2], Yan Wang[1,2], Ruichao Li[1,2]

[1]State Key Laboratory of Numerical Modeling for Atmospheric Sciences and Geophysical Fluid Dynamics, Institute of Atmospheric Physics, Chinese Academy of Sciences, Beijing 100029, China

[2]College of Earth and Planetary Sciences, University of Chinese Academy of Sciences, Beijing 100049, China

[3]School of Software Engineering, Chengdu University of Information Technology, Chengdu 610225, China

*Corresponding author: Zhenghui Xie (zxie@lasg.iap.ac.cn)

**Abstract.** Located in the mountainous area of southwest China, the Chengdu-Chongqing Urban Agglomeration (CCUA) was rapidly urbanized in the last four decades, has led to a three-fold urban area expansion, thereby affecting the weather and climate. To investigate the urbanization effects on the thermal environment in the CCUA under the complex terrain, we conducted the simulations using the advanced Weather Research and Forecasting (WRF V4.1.5) model together with the combining land-use scenarios and terrain conditions. We observed that the WRF model reproduces the general synoptic summer weather pattern, particularly for the thermal environment. It was shown that the expansion of the urban area changed the underlying surface's thermal properties, leading to the urban heat island effect, enhanced by the complex terrain further. The simulation with the future scenario shows that the implementation of idealized measures including returning farmland to forests, expanding rivers and lakes can reduce the urban heat island effect and regulate the urban ecosystem. Therefore, the urban planning policy can have potential to provide feasible suggestions to best manage the thermal environment of the future city toward improving the livelihood of the people in the environment.

## 1. Introduction

With urban area expansion, the CCUA's lower surface has changed compared with the natural land surface, leading to the heat island effect, which is also an important factor in global warming(Kalnay et al., 2003; Kawashima, 1975; Ning et al., 2019). As one of the most dramatic land-use changes, urbanization alters land surface physical properties, including albedo, emissivity, heat conductivity, and morphology, making urban areas exhibit greater heat capacity, Bowen ratio, and roughness(Robaa, 2011). The impermeability of urban land surface reduces water vapor evaporation and increases sensible surface heat. The multiple reflection and absorption of radiation in the urban canopy make the energy absorbed by a city in the daytime more difficult to dissipate in the form of long-wave radiation at night. These changes in land surface characteristics significantly affect the surface energy budget, boundary layer

height (PBLH), thermal structure, and local/regional atmospheric circulation(Kawashima, 1975; Oke, 1995; Berling-Wolff et al., 2004; Hamdi et al., 2010; Hamdi et al., 2010;) . Therefore, it is imperative to assess the changes in urbanization and develop adaptation strategies.

Numerical simulation has been used to investigate the urban heat island effects on cities, such as Tokyo, Phoenix Metropolitan, Beijing, and Hangzhou(Berling-Wolff et al., 2004; Chen et al., 2014; Saitoh et al., 1996 ; Wang et al., 2020 ). Urbanization also changes a city's precipitation by enhancing the spatial heterogeneity of the rainfall or making it extreme (Yang et al., 2019). Urbanization of cities in the arid and semi-arid areas can cause pronounced urban drying (Robaa, 2011). For cities under complex terrain conditions, their weather and climate are often exacerbated by the interaction of complex terrain with urbanization (Ning et al., 2018; Yang et al., 2019). On the other hand, cities will face severe water and heat stress (Zhao et al., 2021). Therefore, the demand for a suitable plan to alleviate stress is exigent.

In this study, we investigated the interaction of complex terrain and urbanization on the thermal environment of the urban agglomeration for CCUA, located in the mountainous area of southwest China. We further researched the effects of land-use planning policies on the heat stress of the urban area. This research (1) clarifies the urban warming pattern caused by the urban expansion of the CCUA in the past, (2) measures the combined impact of complex terrain and urbanization on the summer urban thermal environment, and (3) reveals the potential of implementing the measures of returning farmland to forest and grassland and expanding the area of rivers and lakes to alleviate heat stress in the CCUA.

## 2: Data and methodology

### 2.1 Weather Research and Forecasting (WRF) configuration and study area

We used a numerical model WRF-ARW v4.1.5 (Skamarock et al., 2008), coupled with a single-layer urban canopy model (SLUCM) and the Noah land surface model (Noah LSM, Niu, G.Y., 2011; Z.L. Yang,G.Y., 2011,), to study the impact of urbanization on the regional thermal environment. We chose CCUA as the study area, set up three one-way nested domains in the horizontal direction (Fig. 1a), with resolutions of 1km, 5km, 25km respectively, and divide the atmosphere into 32 vertical layers. July, the hottest month in 2018, was selected as the simulation period, and the first 48 h of the simulation results was discarded as the spin-up time of the model. The forced initial field data simulated in the model was the re-analyzed data of operational global analysis and forecast data, which are on 0.25×0.25-degree grids, prepared operationally every six hours, from National Centers for Environmental Prediction (NCEP, https://rda.ucar.edu/datasets/ds083.3). The main physical schemes for the model selection are the following: the microphysics scheme as the Thompson et al. Scheme; the long-wave radiation scheme as the RRTM scheme ( Mlawer, E.J.,1997 ); the shortwave radiation scheme as the Dudhia scheme ( Dudhia, 1989 ); the near-ground layer and boundary layer schemes as the Revised MM5 surface layer scheme ( Monin and Obukhov, 1959) ; the land surface scheme as the Noah Land Surface Model; the planetary boundary layer as the BouLac PBL (Hong et al., 2006) ; As for how to select the parameterization schemes above of WRF model, we refers to the previous research of the cooling efficiency of adaptation

strategy in the Chengdu Chongqing metropolitan region, the scheme adopted by them verifies that SLUCM and other physical parameterization schemes have good adaptability (Liu et al., 2018).

The urban environmental conditions of CCUA are similar, depending on the same comprehensive transportation network, with two megacities (Chengdu (CD) and Chongqing (CQ)) as the core city, and the other 14 smaller cities distributed among them, thus forming a large urban agglomeration(Wang et al., 2015 ; Xiaojuan et al., 2018 ). The CCUA is located in the Sichuan Basin in the central and southern Asian continent (between latitude 28"10' and 32.25'), surrounded by the Qinghai-Tibet Plateau, Daba Mountain, Huaying Mountain, and Yungui Plateau(Wang et al., 2015). The surrounding mountains are primarily between 1,000 and 3,000 meters above sea level. Compared with plateaus and plains, the topography is overly complicated. The Sichuan Basin is relatively humid, situated in the mid-subtropical zone, and it also has marine climate characteristics(Richardson, 2008). In winter, the temperature in this area is the highest at that latitude. This is due to the occlusion of the terrain, and the rich cold air is blocked by the surrounding plateaus and mountains(Yuan and Xie, 2012). The annual precipitation in the Sichuan Basin is 1,000 to 1,300 mm(Shao et al., 2005). The mountain areas on the western edge of the basin have higher annual precipitations (1,500–1,800 mm), a prominent rainy area in China(Yuan et al., 2012).

## 2.2 Land-use change and land-cover datasets

The underlying surface influences factors, such as soil thermal conductivity, vegetation impedance, reflectivity, roughness, and thermal inertia. The thermal inertia, in turn, affects the boundary layer structure and the land surface process. Therefore, more refined underlying surface information will improve the model simulation effect significantly.

WRF has two default land-use dataset types: the Advanced Very High-Resolution Radiometers from the US Geological Survey (USGS) and Moderate-resolution imaging spectroradiometer (MODIS). The acquisition time of USGS data was from April 1992 to March 1993, while MODIS's latest land-cover data was in 2010. The default data accuracy is low, and the timeliness is not enough, restricting the simulation accuracy of the model. Therefore, we replaced the WRF default data with the more accurate land use data. We get the 30-meter spatial resolution fusion land cover data for the two periods (viz. 1980, the Historical Scenario (Fig. 2a) and 2018, the Urban Scenario (Fig. 2b)) from the Institute of Geographic Sciences and Natural Resources of the CAS (Resource and Environment Science and Data Center https://www.resdc.cn/data.aspx?DATAID=264/). They were developed and verified in detail from medium-resolution satellite images. We reclassified the land use and land cover data (LUCs) into 24 categories according to the International Geosphere-Biosphere Project classification scheme (USGS, 2003), to meet the classification standard in WRF model. The impervious surfaces of towns, industrial and mining lands, and roads were integrated as urban land-use types. The methods and criteria for reclassification are shown in Table 1.

After statistically calculating the land cover, it is found that from 1980 to 2018, the growth of urban land

use types in the CCUA increased to nearly 9,000 square kilometers, which shrank the area of dry land, paddy field, and water wetland. According to the more accurate land use data in 1980 and 2018, urban land-use type is larger and can better reflect the actual situation of the underlying surface of the CCUA.

To explore the impact of urbanization on the thermal environment of the CCUA based on the 2018 land use and land cover dataset, we replaced the urban land type with the nearest natural land use type of the patch. Here, we term the land data set without city type "No Urban Scenario (Fig. 2c) ".

In addition to the three land-use scenario data, we also planned and designed a Future Scenario land use of urban agglomeration landscape to explore the mitigation effect of landscape planning on the thermal environment stress of urban agglomeration. Urban agglomerations are brand-new regional units that have emerged from industrialization and urbanization into a higher stage (Bruinsma and Rietveld, ; Jun et al., 2012 ; Kawashima, 1975 ; Strange, 2008 ). The CCUA is the most vitality region with a high potential for economic development in southwest China (Wang et al., 2015;). However, the urban agglomeration is an extremely sensitive area where a series of ecological and environmental problems are highly concentrated and intensified (Bruinsma and Rietveld,). For such ecological environment pressure, we can design and plan ecological corridors and ecological barriers in landscape ecology according to the natural geography, vegetation ecology, water system, and topography of urban agglomeration. It is expected that these ecological corridors and barriers can alleviate the urban heat stress caused by urbanization, meeting the growing cultural demands of the people. Based on the land-use dataset of the Urban Scenario (Fig.2b), we designed the ideal land use and land cover scenarios in the future, the Future Scenario (Fig. 2d). The specific method is to return farmland to forest and grassland in the five ecological protection areas around the urban agglomeration. Eight land ecological corridors and seven water system corridors were designed according to the hills, mountains, and water systems in the urban agglomeration. In the corridor, the farmland should be returned to grassland to expand the river lake wetland. We plan the land use of the future urban agglomeration according to the government's planning documents (see Section 2.3 for details).

All the four land-use datasets with a 30-meter resolution were resampled for 1 km as the underlying surface data of the model in the simulated area.

**2.3 Planning and designing Future Scenario of CCUA**

We will present how to plan and design the "Future Scenario" of CCUA in Fig.2d mentioned above in this subsection. We first divided the scope of ecological protection areas and ecological protection barriers in future land use planning (Fig.3). The designing and planning Future Scenario is based on the two planning documents issued by the government: "Chengdu Chongqing Urban Agglomeration Development plan, 2014-2020", which was issued by Development and Reform Commission of The People's Republic of China (PRC), Ministry of Housing and Urban-Rural Development of PRC (https://www.ndrc.gov.cn/fzggw/jgsj/ghs/sjdt/201605/W02019101064289584_2500.pdf), and the other planning document, "Guidelines for Ecological Protection and Restoration Project of Mountains, Rivers,

Forests, Fields, Lakes and Grasse,2020 (trial version)", which was issued by three other government departments: Ministry of Natural Resources of PRC, Ministry of Ecological Environment of PRC, and Ministry of Finance of PRC (https://www.cgs.gov.cn/tzgg/tzgg/202009/W020200921635208145062.pdf). These two planning documents point out a series of ecological and environmental protection measures such as returning farmland to forests and grasses and expanding lake water system wetlands.

According to the "Chengdu Chongqing Urban Agglomeration Development Plan,2014-2020", we define the location of the ecological protection areas of the urban agglomeration according to the guidance of the government, including the five nature reserves (Sichuan-Yunnan Forest Reserve, Qinba Biodiversity Ecological Function Zone, Da-Xiao-Liang Mountain Water and Soil Conservation Ecological Function Zone, Wuling Mountain Ecological Diversity and Soil and Water Conservation Ecological Function Zone, and Three Gorges Reservoir Water and Soil Conservation Ecological Function Zone) around the urban agglomeration and the land ecological corridor and water ecological corridor within the urban agglomeration. According to the document "Guidelines for Ecological Protection and Restoration of Mountains, Rivers, Forests, Fields, Lakes and Grasses, 2020 (trial version)", we replace 75% of farmland in 5 nature reserves with mixed grassland / shrubland. In the land ecological corridor, we replace 60% of the farmland with cropland / Woodland mosaic, and in the water ecological corridor, we replace the farmland within one kilometer along the river with wetland.

By comparing the Future Scenarios land use and the Urban Scenarios land use, we expect that the planned landscapes will improve the thermal environment of Urban Agglomerations in the summer, enhancing the living comfort in the urban agglomerations.

## 2.4 Experimental design

To study the impact of urbanization on the surrounding environmental and meteorological elements under complex terrain, seven experiments were designed as described in Table 2: five types of land use scenarios ("WRF Default Scenario", "History Scenario", "Urban Scenario", "No-Urban Scenario", and "Future Scenario") and two type of terrain conditions ("Topography (Fig. 1b)", the current situation of the original complex terrain of the CCUA; "No-Topography (Fig. 1c)", that is smoothing the mountainous terrain around the CCUA). The high-altitude mountains around CCUA have been removed through multiplying the altitude of the high-altitude area in the study area by a certain proportion of the scaling factor which makes the terrain of the whole simulation area smoother. The physical parameterization schemes and simulation time periods, as well as the study area, were the same as those mentioned in the previous section, except the underlying surface land use datasets and terrain changes. Fig.1 (d) is the distribution of more than 2000 meteorological observations stations from the China Meteorological Administration.

First, to verify the results of "Urban Scenario" and "Topography" (exp. 5), the land use data-driven model can reproduce the weather conditions in July 2018 more accurately than the "WRF Default Scenario" (2010 MODIS) vs. "Topography" (exp. 1). We compared the experimental results of exp. 5 and exp. 1

with the observation data of the National Meteorological Information Center of China Meteorological Administration. Secondly, by subtracting the results of "No-Urban Scenario" and "No-Topography" (exp. 4) from "Urban Scenario" and "No-Topography" (exp. 3), we compared the influence of the single urbanization factor on the thermal environment of urban agglomeration. Then, the result of exp. 5 minus exp.3 was taken as a single terrain factor affecting the thermal environment.

Topography directly affects the local atmospheric circulation. Here is a research question: will topography interact with urbanization to jointly affect the thermal environment of urban agglomerations? We refer to Yang's method to explore the interaction between this terrain and urbanization before we quantified the impact of urbanization on the thermal environment of urban agglomeration under complex terrain, i.e.,[ exp. 5 - (exp. 3 + exp. 6) + exp. 4 ] (Yang et al., 2019 ) . By comparing exp. 5 and exp. 2, we quantified the impact of urbanization on the thermal environment under complex terrain. Finally, by comparing the land-use of Future scenarios after planning (exp. 7) with the current situation (exp. 5), we can explore whether landscape planning inside and outside the urban agglomeration can alleviate urban heat stress.

**3 Results and discussion**

**3.1 Model verification**

To verify the model performance, we compared the simulated monthly and daily mean spatial temperature models in July 2018 with the WRF default land-use dataset and with observations of the meteorological stations (Fig. 4a, 4c, and 4e). The correlation coefficient matrix of the 2018 Land use simulation results have a high spatial correlation coefficient ($p \leq 0.05$) over the whole the CCUA region (Fig. 5), Ten variables' correlation coefficient was calculated, such as the surface skin temperature (TSK)、2 m air temperature (T2)、ground heat flux (GRDFLX)、upward heat flux at the surface (HFX)、latent heat flux at the surface (LH)、 downward latent heat flux at the surface (LW_dw)、upward latent heat flux at the surface (LW_up)、shortwave(SW)、PBLH、net radiation (Rn). The rationality of the correlation coefficient matrix shows that the configuration of the model is reasonable. However, the simulation underestimated the surface 2m air temperature by 0.75~2.5 °C compared with observation (Fig. 4b). Previous studies had reported similar bias, characteristic of the WRF model (Wang et al., 2015; Wang et al., 2013), with most overestimations occurring in the urban areas of the respective study areas.

Furthermore, the model exhibited a large negative deviation in the mountainous area around the CCUA (Fig. 4b) due to the systematic error of underestimating wind speed and temperature in the area. The Root-Mean-Square-Error between the observed and exp.5 (Urban Scenario & Topography) simulation results was about $-2 \sim 4$ °C (Fig. 4d). The spatial distribution of the correlation coefficient between the observation and the monthly mean 2 m temperature simulated by exp.5 is shown in Fig. 4f. It has a high correlation in the whole urban agglomeration. Since we are concerned with the air temperature changes caused by the land cover and terrain interaction of different underlying surfaces, some systematic

deviations can be offset by the sensitivity experiments rather than the accurate reproduction of the absolute temperature in the study area.

## 3.2 The interaction of complex terrain and urban expansion

### 3.2.1 Topography effect

To explore the impact of terrain on the thermal environment of urban agglomerations in the summer, we made the land use of the CCUA constant (No-Urban Scenario), compared the results under the two different terrain scenarios: original complex terrain (Topography) and smoothed topography (No-Topography). Comparing Fig. 6a with Fig. 6b, we can see that the influence of terrain on the temperature pattern is basically consistent with the current distribution pattern of summer monthly average

temperature, and terrain is the main factor determining the temperature pattern. In Fig. 6b, the temperature inside the CCUA (compared with flat terrain) increases by about 10 ℃ when there is complex terrain. Simultaneously, the temperature in the plateau and mountainous areas around the CCUA decreases by more than 10 ℃. The complex terrain would form a lower temperature plateau mountain climate than the smooth plain terrain.

Sichuan Basin, where the CCUA is located, has a concave landform. The closed topography leads to low wind speed in Sichuan Basin, making the heat in the basin difficult to dissipate. Therefore, the thermal environment here is more severe than that in the flat terrain. In Fig. 7c, 7g, and 7k, we observed that the complex terrain would increase the HFX and LH of the urban agglomeration. At the same time, due to the high altitude around the basin itself, it will significantly raise the atmospheric boundary layer of the

urban agglomeration. These are the crucial attributions for the temperature rise caused by the complex terrain.

    The many rivers in Sichuan Basin make the southeast monsoon convey large quantities of water vapor, blocked by the mountains around Sichuan Basin. The southeast of the mountainous area is low, which is favorable for receiving water vapor. On the contrary, the northwest Mountainous area is of relatively high

altitude, thereby conducive to water vapor loss, causing increased air humidity. Therefore, the topography is pertinent to forming a humid and hot climate in summer in the CCUA.

### 3.2.2 Urban effect

    To determine the summer warming caused by a single urban land expansion factor, we conducted two groups of experiments: exp. 3 and exp. 4. Both groups of experiments smoothed the terrain to eliminate

the influence of complex terrain. We observed that the urban expansion would cause the temperature of the whole urban agglomeration region to increase by ≈ 0.8 ℃ (Fig. 6c). Especially in the core areas of urban agglomeration (CD and CQ), the temperature increased significantly, nearing 1.0 ℃ increase in the main urban area. The temperature inside the urban agglomeration was considerably higher than on the outside, attesting to the "heat island effect."

Urbanization will significantly change the surface albedo, heat capacity, and thermal conductivity of the

underlying urban surface. The change of surface heat flow caused by urbanization is shown in Fig. 7c. The urban impervious surface absorbs more shortwave radiation downward, and the GRDFLX is stored more in the daytime and released more at night. Daytime surface temperature is mainly due to the increase of urban surface HFX, with a maximum increase of 90 W/m$^2$ in the core urban area of CD and CQ (Fig. 7b). The HFX rise directly elevates the near-surface temperature. Due to the impervious city surface, the city's evapotranspiration was lower than that of the suburb, and the LH during the day was significantly reduced, the maximum reduction can reach 110 W/m$^2$ (Fig. 7f). The warming effect caused by urbanization enhances the turbulence and increases the PBLH (Lin et al., 2008). Also, the variation area and high-value area of the PBLH (Fig. 7i, 7j) are the temperature variation area, concentrated in the core urban area. Here, the urban PBLH increased by 40~170 m. Therefore, under the same external meteorological conditions, the HFX of urban construction land was higher, the LH was lower, and Bowen ratio was higher when compared to the vegetation coverage area around the city. The HFX and Bowen ratio of the urban surface were significantly higher than those of surrounding vegetation. This occurrence increases the heating of the urban surface into the lower atmosphere, an important mechanism of the urban heat island effect.

### 3.2.3 Urban-Topography interaction

By comparing the results of exp. 3, exp. 4, exp. 5, and exp. 6, we conclude terrain is the most crucial factor in forming local weather and climate pattern. In the case of complex terrain and urbanization, the terrain would affect the weather and climate simultaneously, causing climate change (Fig. 6d, Fig. 7d, Fig. 7h, and Fig. 7i). Compared with the urban warming effect (caused by a single urbanization factor), the warming effect of the urban core area is more evident after the complex terrain is added. The warming areas are more concentrated in the urban core area. The average temperature in the core areas of CD and CQ increases by more than 1.5 ℃.

Due to the joint influence of topography and urban expansion, the HFX increased by about 30 W/ m$^2$. the LH increased by 30~60 W/ m$^2$, extensively in the southwest of the CCUA, while the LH in the northeast of CCUA decreased by 20 W/m$^2$. The boundary layer of the city has been raised significantly, especially in the main urban areas of CD & CQ, with the highest elevation of 180 m.

Considering the heat flux, temperature, boundary layer and other factors mentioned above, we think that the topography further enhances the heat island effect in the CCUA and the urban core area of CD&CQ.

### 3.3 Historical warming of urban agglomerations

In front, we quantitatively studied the influence of different factors on the urban thermal environment, such as single urban factor, single topography factor and the combined influence of urban and topography factors. The results reveal in detail the mechanism of how these factors affect the urban thermal environment.

To determine the summer warming caused by historical urban land expansion, we calculated the

simulated 2-m air temperature difference between the urbanized land use in 2018 and the historical land use before urbanization in 1980 (Fig. 8a~8c) by the exp.1 & exp.5. The whole region experienced some warming, with notable ones occurring in areas consistent with the location of the urban grid in 2018. At the same time, the northeast part of the urban agglomerations was warmed, probably resulting from the urban agglomerations effect or terrain hindering heat dissipation. The monthly average temperature of 2-m air in July of 2018 was 0.75 °C higher than that of 1980. The most significant temperature increase occurred in the main urban area of CD and CQ. The maximum temperature rise reached 0.8 °C. Note that the anthropogenic heat was not considered in the simulation process of this study. Therefore, the simulated warming is attributed to the increase of urban land only. Fig. 8d~8f shows the frequency distribution of the heating amplitude of all grid points of CCUA, the warming range of CD urban area is between 0.5 °C and 1.1 °C, while that of CQ urban area is between 0.4 °C and 0.8 °C.

Fig. 10a, Fig. 10b, and Fig. 10c show the diurnal variation of surface temperature, 2-m air temperature, and PBLH in the summer of the CCUA, CD, and CQ in 1980 and 2018, respectively. Here, regardless of the CCUA, CD, or CQ, the daily average surface temperature and the temperature of 2-m air simulated by the 2018 urbanization scenario were significantly higher than those of the 1980 historical scenario, and the daily average atmospheric boundary layer represented by the histogram is also increased by 50~100 m.

Compared with Fig. 11a and 11b, the change of surface radiation balance caused by urbanization was evident. The impervious surface layer of the CCUA absorbed more downward shortwave radiation, and the GRDFLX storage was larger during the day, while the GRDFLX released was larger at night. The surface temperature increased mainly through the HFX of the urban surface before gradually relieving from the HFX. The HFX reduction directly elevates near-surface temperature. Due to the decrease in soil evapotranspiration, the LH decreased significantly. From 1980 to 2018, large cities exhibited reduced soil moisture and near-surface wind speed, resulting in a lowered evaporation. The higher the surface temperature, the more intense was the long-wave radiation, and the higher the net radiation energy lost in the daytime. Similar scenarios ensued for the main urban areas of CD and CQ because the proportion of impervious surface was higher. Changes in the surface heat flux became more apparent (Fig. 11c, Fig. 11d, Fig. 11e, and Fig. 11f).

The two columns on the left side of Fig. 12 show the components of surface heat flux during the simulation period caused by urbanization. The upper and lower endpoints of the box graph represent the maximum and minimum values of the heat flux, while the middle, upper side, and lower side represent the average, 1/4, and 3/4 quantiles, respectively. The four boxes from left to right in each subgraph represent the ground heat flux Surface heat flux, HFX, LH, and Rn. The scatter points and curves on the right side of the box represent the distribution of surface heat flux values. Thus, it is easy to affirm that the mean and maximum values of HFX and Rn increased from 1980 to 2018. Similarly, in CD and CQ, the impervious surface area was higher, increasing significantly.

Therefore, we suggest that urbanization will inevitably produce a heat island effect in urban

agglomerations, especially under a complex terrain. Consequently, some effective and ideal measures could be adopted to alleviate the heat island effect caused by urbanization.

**3.4 Mitigation of future heat stress**

Due to the significant urban heat island effect associated with urbanization, to explore reasonable measures to alleviate the urban heat island effect and improve the living comfort of urban residents, we designed the future land-use scenarios. To explore the extent to which the urban heat island effect can be alleviated by returning farmland to forest and grassland and expanding the river lake wetland area of urban agglomeration, we compared the future scenario with the current urban scenario (Fig. 9). In the central part of the CCUA, the average temperature dropped by about 0.5 °C in summer in July and decreased at night and during the day. The primary cooling interval was ≈0.4~1 °C and the cooling rate in CQ was higher than that in CD. This situation occurred because that the planned land use is closer to the urban area of CQ than to the urban area of CD. Therefore, the response of CQ to this measure was more obvious in landscape planning.

Comparing the Future scenario (Fig. 10) with the 2018 Urban scenario, it was tedious to find planning measures to reduce the air temperature and ground temperature 2 m above the CCUA area, CQ area, and CD area. After the planning, the overall average temperature dropped by 0.2~0.67°C, while the daily average PBLH of CCUA, CD, CQ, and other cities dropped 50~150 m in July. Most of the days, the decline was > 100 m.

Comparing the two columns on the right side (Fig.12), the city's heat flux changed after the urban agglomeration planning. With the urbanization scenario in 2018, the average GRDFLX decreased by about 30 W/m$^2$, the maximum HFX decreased by 38 W/m$^2$, the LH increased by 47 W/m$^2$, and the average Rn increased by 14 W/m$^2$.

There is a significant difference between the urban heat budget and the natural underlay surface. From the results, we observed that the planning measures have significantly improved the thermal environment of the city because the planned land use increases the natural underlay surface of vegetation and water system. Urban impervious surface and building surface evinced higher surface temperature, which was decisive in the HFX. Whereas, the natural underlay surface dominated the LH, caused by transpiration and evaporation of vegetation and wetland. Because the surface temperature was lower than the impervious surface of the city, planning policy can reduce the urban heat island effect and improve the comfort of human settlements.

**4. Conclusions and discussion**

In this study, we investigated the CCUA summer urban warming effect and its adaptation strategies under the complex terrain in Southwest China through conducting seven simulations using WRF/SLUCM model with the combined five land-use scenarios, including (History, Urban, No-Urban, and Future Planning Scenarios), and two kinds of terrain (original complex terrain and smoothed terrain). It was

found that urban land-use types of the CCUA increased to nearly 9,000 square kilometers. The simulations using 2018 land-use data and original complex terrain showed that the WRF model reproduces a general pattern of summer weather against the observed temperature. In the past 40 years, the changed underlying surface led to the urban heat island effect, increasing the urban temperature by 0.75 °C. The impervious surface absorbed more Rn and stored the energy in the buildings and pavement. The remainder transmitted the HFX to the air through the turbulence exchange, and the HFX rose by > 90 W/m$^2$, raising the boundary layer. In addition, the transpiration and evaporation from the urban underlay surface decreased, leading to lowering LH by ≈110 W/m$^2$. These energy balance changes eventually led to temperature rise, and finally to the urban heat island effect. Moreover, the mountainous area around Sichuan Basin is complex in topography, making it difficult for heat to diffuse. This scenario further strengthened the urban heat island effect, enhancing it by ≈30%.

The simulation for the future planning scenario shows that the implementation of idealized measures (such as returning farmland to forest and river lake expansion) can reduce the urban heat island effect. Likewise, it can regulate the urban ecosystem, such as the average temperature of 2 meters in the summer of urban agglomeration decreased by ≈0.2~0.67 °C. Also, the average net radiation on the surface was reduced by 17 W/m$^2$. Finally, we anticipate that urban planning policy can provide effective suggestions for future urban thermal environment management and improve living comfort.

This study focuses on exploring the impact of urbanization in complex terrain environment on local geothermal environment using the WRF model with the USGS data of WRF's default data, and the IGSNRR land use data that is more timely and more suitable for CCUA in the local research area. We may also discuss simulation uncertainties from other land use and land cover data in future.

*Acknowledgments*

This work was supported by the Strategic Priority Research Program of Chinese Academy of Sciences [Grant number: XDA23090102], the National Natural Science Foundation of China (NSFC) project [Grant number: 41830967], the National Meteorological Information Center, China Meteorological Administration for data support. We also thank the editor, Dr. Gabriele Messori, and the reviewer, A.P. Hideki Takebayashi, and the two anonymous reviewers for their kind comments for the manuscript

Table1 Rules for converting the classification standard of LUCs from IGSNRR to USGS

| IGSNRR-Level 1 | IGSNRR–Level 2 | | IGSNRR(ID) -> USGS(ID) | USGS |
|---|---|---|---|---|
| 1- Cropland | 11- Paddy field | | 11 -> 3 | 3 - Irrigated cropland and pasture |
| | 12- Dry land | | 12 -> 2 | 2 - Dry land cropland and pasture |
| 2- Woodland | 21- Woodland | | 21 -> 11 | 11- Deciduous broadleaf forest |
| | 22- Shrubland | | 22 -> 8 | 8 - Shrubland |
| | 23- Open woodland | | 23 -> 19 | 19- Barren or sparsely vegetated |
| | 24- Other woodlands | | 24 -> 15 | 15- Mixed forest |
| 3- Grassland | 31- High coverage grassland | | 31 -> 7 | 7 - Grassland |
| | 32- Medium coverage grassland | | 32 -> 19 | 19- Barren or sparsely vegetated |
| | 33- Low coverage grassland | | 33 -> 9 | 9 - Mixed grassland /shrubland |
| 4- Water bodies | 41- Canal | $6≤N1≤9;$ $(N2\sim N4)≤1$ | 41 -> 16 | 16- Water bodies |
| | 42- Lake | | 42 -> 16 | |
| | 43- Reservoir/Pit/Pond | | 43 -> 16 | |
| | 44- Permanent glacier and snow | | 44 -> 24 | 24- Snow or ice |
| | 45- Tidal flat | | 45 -> 17 | 17- Herbaceous wetland |
| | 46- Beaches of rivers and lakes | | 46 -> 17 | |
| 5- Urban and built-up land | 51- Urban and built-up land | | 51 -> 1 | 1 - Urban and built-up land |
| | 52- Rural Settlements | | 52 -> 1 | |
| | 53- Other construction land | | 53 -> 1 | |
| 6- Unused land | 61- Sand | | 61 -> 19 | 19- Barren or sparsely vegetated |
| | 62- Gobi Desert | | 62 -> 19 | |
| | 63- Saline-alkali soil | | 63 -> 19 | |
| | 65- Bare land | | 65 -> 19 | |
| | 66- Bare rock | | 66 -> 19 | |
| | 67- Other unused land | | 67 -> 19 | |
| | 64- Swamp | | 64 -> 17 | 17- Herbaceous wetland |
| Count the types of LUCs of 9 adjacent grids and the number of various LUCs; The number of each LUC type is recorded from large to small: N1, N2, N3... N9, ranging from 9 to 0; "/" : means "And,Or",that is "&,|". | | $6≤(N1+N2)≤9;$ $5≥N1≥N2≥2;$ $(N3\sim N5)≤1$ | 11+12 -> 4 | 4 - Mixed dryland /irrigated cropland and pasture |
| | | | (11/12) + (31/32/33) -> 5 | 5 - Cropland /grassland mosaic |
| | | | (11/12 )+ (21/22/23/24) -> 6 | 6 - Cropland /woodland mosaic |
| | | | (31/32/33) + 22 -> 9 | 9 - Mixed grassland /shrubland |
| | | | 46+31 -> 17 | 17- Herbaceous wetland |
| | | | 46+21 ->18 | 18- Wooden wetland |
| | | | 67 + (31/32/33) -> 20 | 20- Herbaceous tundra |
| | | | 67 + (21/22/23/24) -> 21 | 21- Wooded tundra |
| | | | 67 + (61/62/63/65) -> 23 | 23- Bare ground tundra |
| | | $6≤(N1+N2+N3)≤9;$ $3≥N1≥N2≥N3≥2;$ $(N4\sim N6)≤1$ | 67 + (21/22/23/24) + (31/32/33) ->22 | 22- Mixed tundra |
| | | | * | 10- Savanna |
| | | | | 12- Deciduous needleleaf forest |
| | | | | 13- Evergreen broadleaf forest |
| | | | | 14- Evergreen needleleaf forest |

Table 2 WRF Experimental Design

| Experimental design | WRF Default Scenario | History Scenario | Urban Scenario | No-Urban. Scenario | Future Scenario |
|---|---|---|---|---|---|
| No-Topography | | | exp.3 | exp.4 | |
| Topography | exp.1 | exp.2 | exp.5 | exp.6 | exp.7 |

Urban effect: exp. 3 – exp. 4

Topography effect: exp. 6 – exp. 4

Urban & Topography interaction: exp. 5 – (exp. 3 + exp. 6) + exp. 4

Historical changes: exp. 5 – exp. 2

Mitigation of future planning: exp. 7 – exp. 5

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

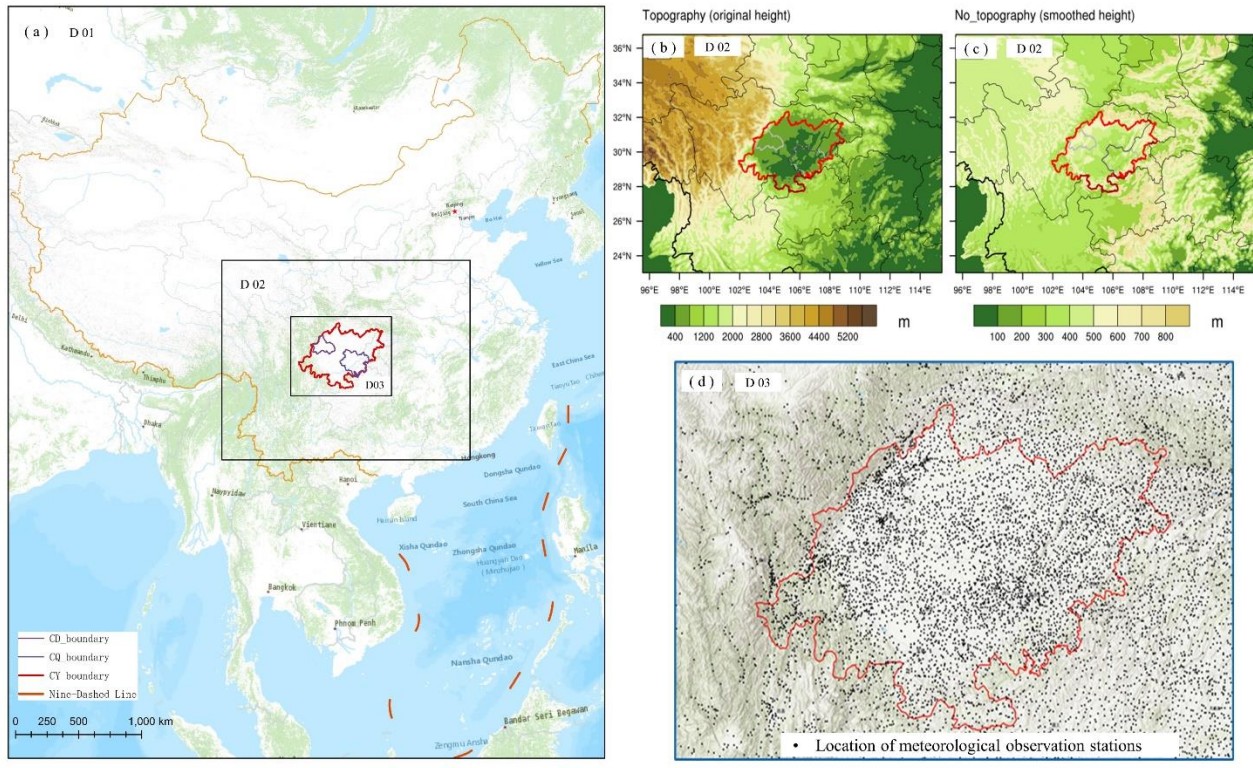

Fig. 1. (a)Configuration of the three nested domains for WRF simulation. (b) The original terrain in the simulated area (m), and (c) after smoothing the terrain. (d) Location of meteorological observation stations.

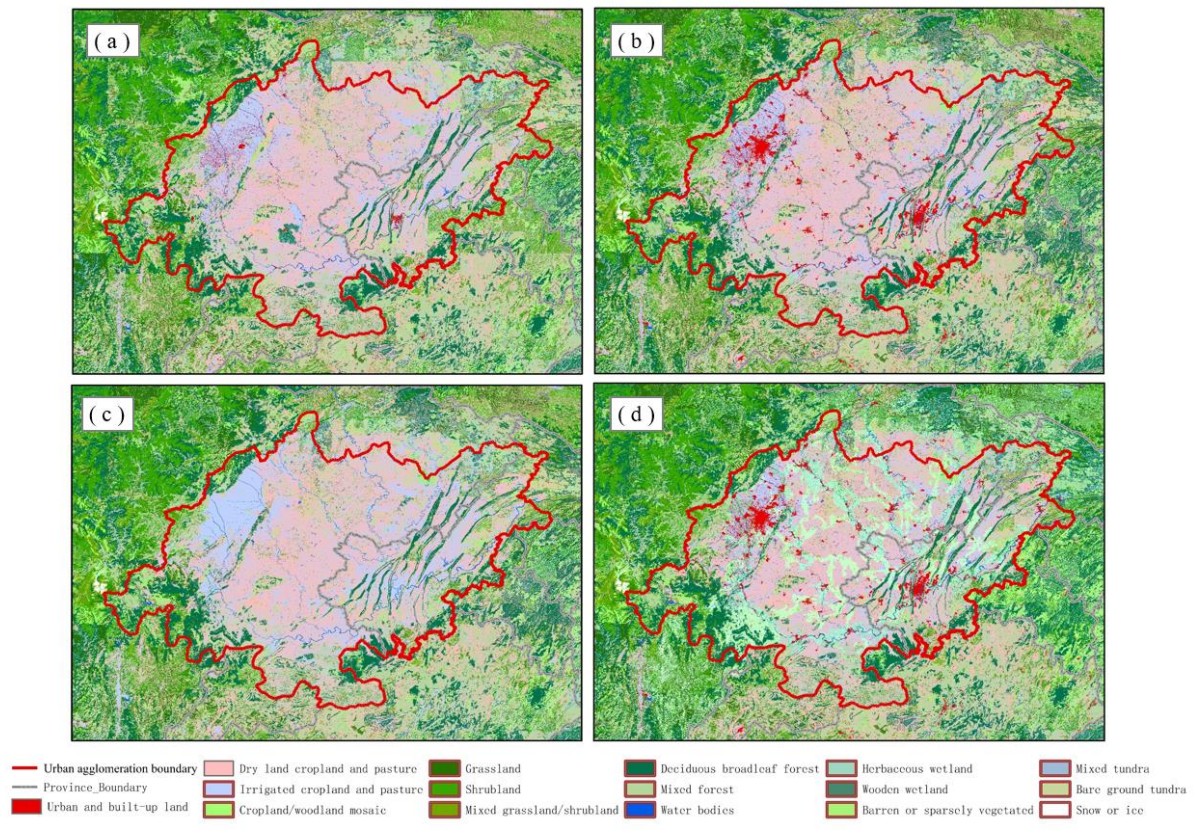

**Fig. 2. Land use/land cover in CCUA: (a) 1980; (b) 2018; (c) No-Urban based on 2018 data; (d) Planned future land use based on 2018 data.**

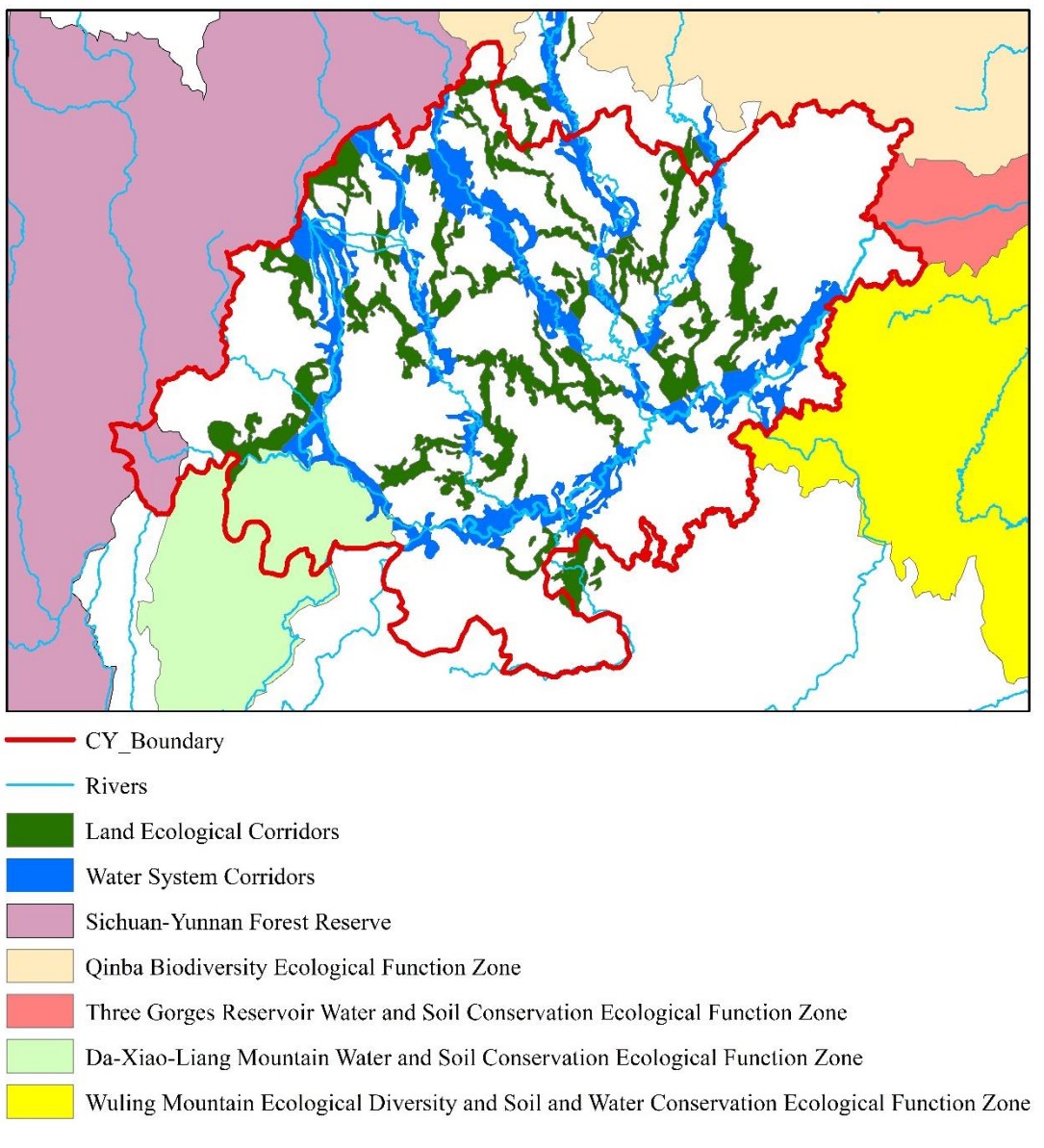

- —— CY_Boundary
- —— Rivers
- ■ Land Ecological Corridors
- ■ Water System Corridors
- ■ Sichuan-Yunnan Forest Reserve
- ■ Qinba Biodiversity Ecological Function Zone
- ■ Three Gorges Reservoir Water and Soil Conservation Ecological Function Zone
- ■ Da-Xiao-Liang Mountain Water and Soil Conservation Ecological Function Zone
- ■ Wuling Mountain Ecological Diversity and Soil and Water Conservation Ecological Function Zone

**Fig. 3 Scope of future land use scenarios planned through government policy documents.**

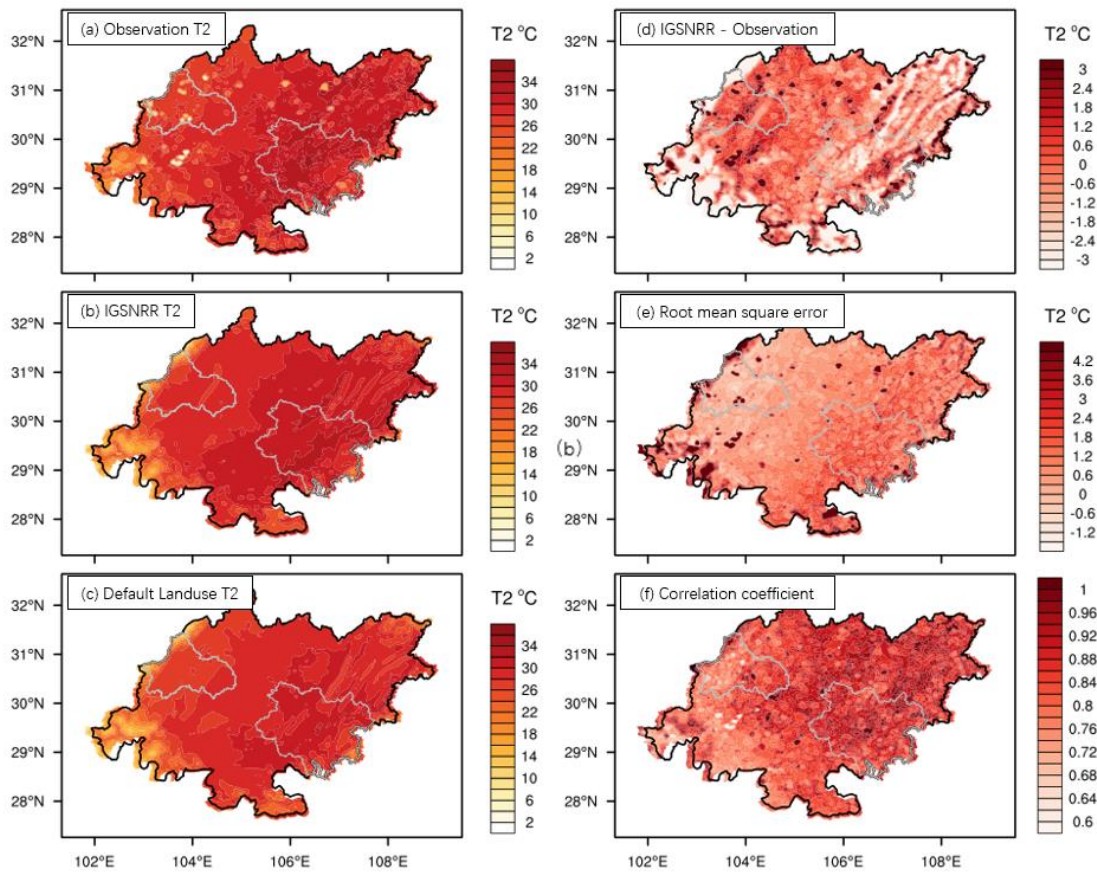

**Fig. 4.** The 2 m-temperature space distribution of (a) observation data from Cimiss, (b) simulation results with 2018 land-use data from IGSNRR, (c) simulation results with land-use data from WRF default USGS 2010 dataset, (d) simulation of IGSNRR minus observation temperature, (e) root mean square error of 2m-temperature between observation and IGSNRR simulation results, and (f) correlation coefficient of 2 m-temperature between observation and simulation.

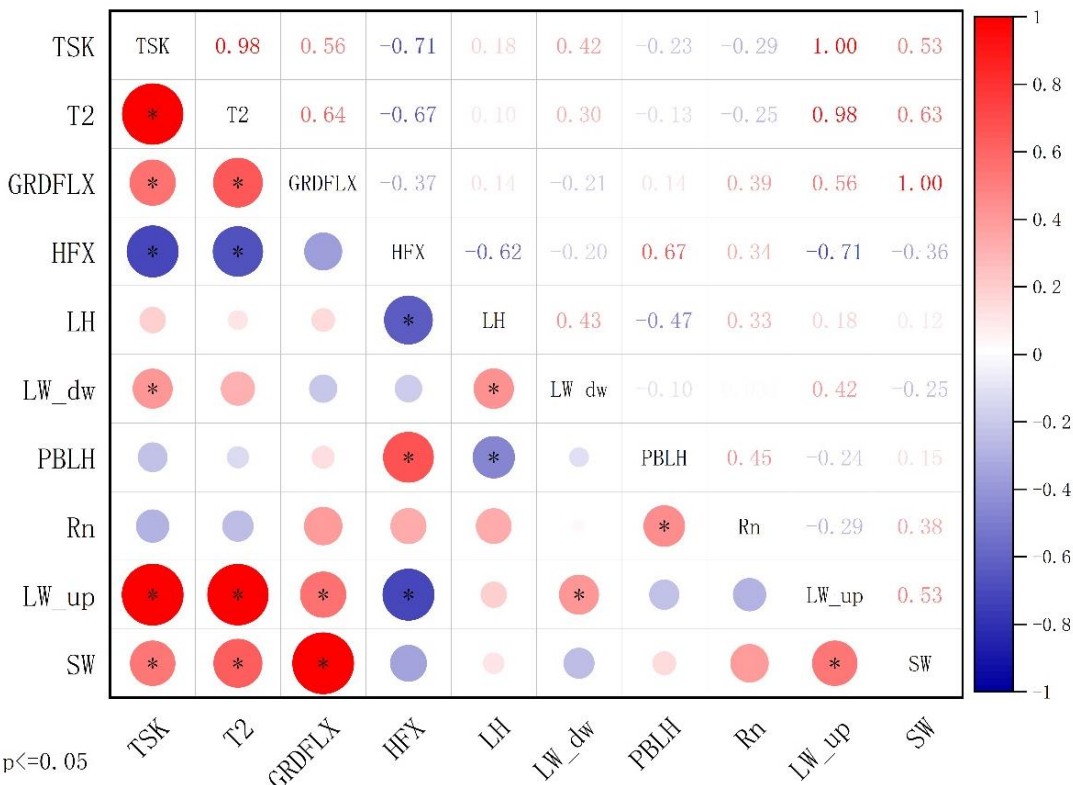

**Fig. 5. Pearson correlation coefficient between ten variable of simulation results by WRF with 2018 land use dataset in CCUA. "*" present the variable was passed significant inspection (p ≤ 0.05).**

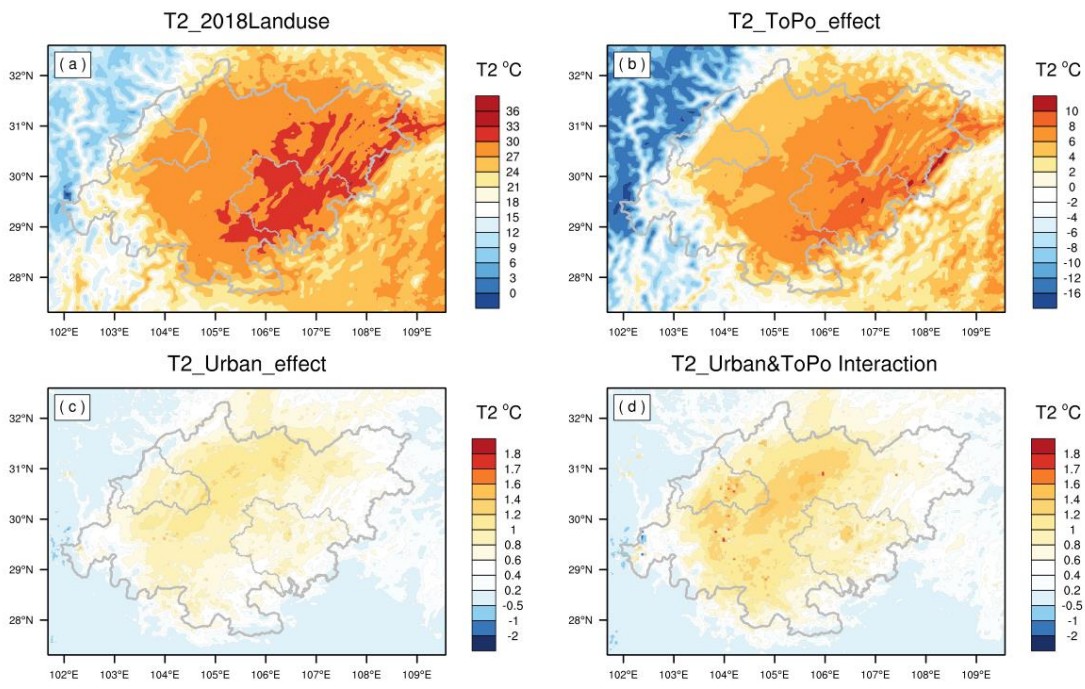

**Fig. 6. Effects of urbanization on 2 m temperature: (a) 2m average temperature distribution with 2018 Land use; (b) Topography effect; (c) Urban effect; (d) Urban-Topography Interaction.**

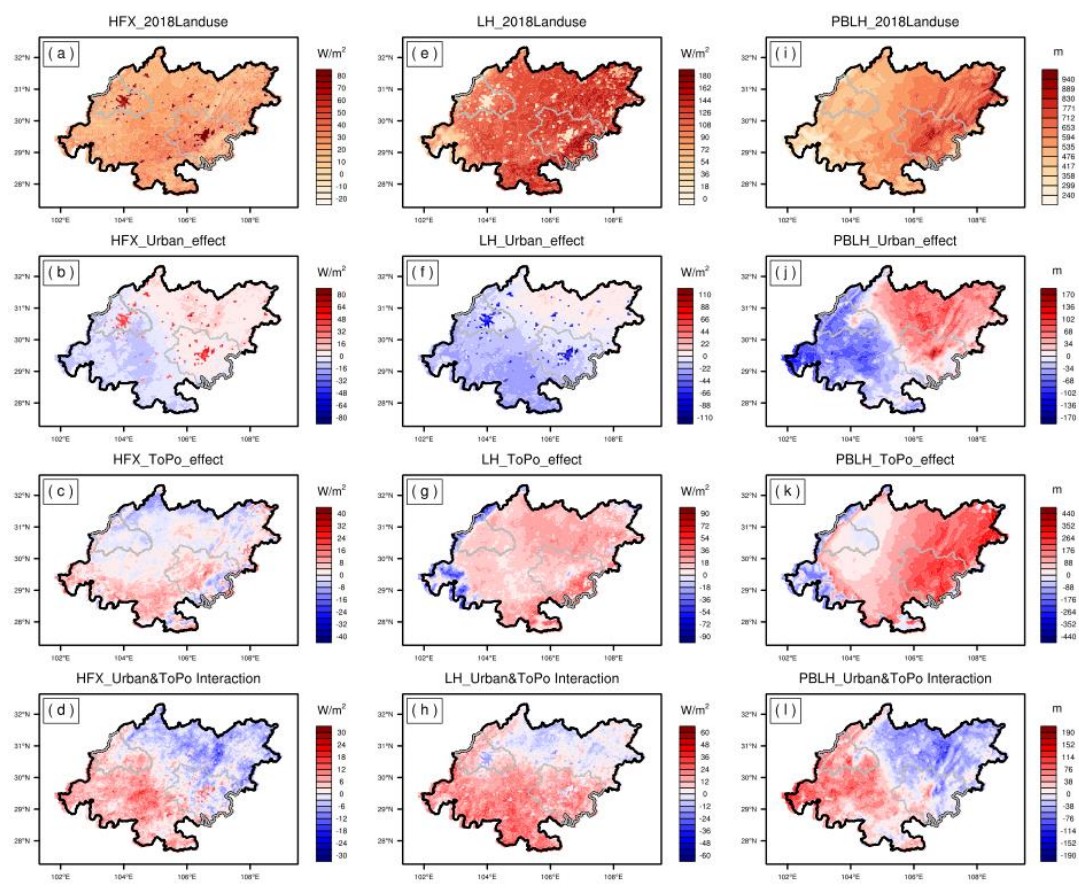

**Fig. 7. Influence of urbanization on urban heat flux and PBLH; (a) 2018 Land use Upward HFX; (b) Influence of urbanization on Urban effect the HFX; (c) Influence of urbanization on urban Topography effect the HFX; (d) Influence of urbanization on urban Urban & Topography Interaction effect the HFX; (e)~(h) Influence of urbanization on urban for LH; (i)~(l) Influence of urbanization on urban PBLH.**

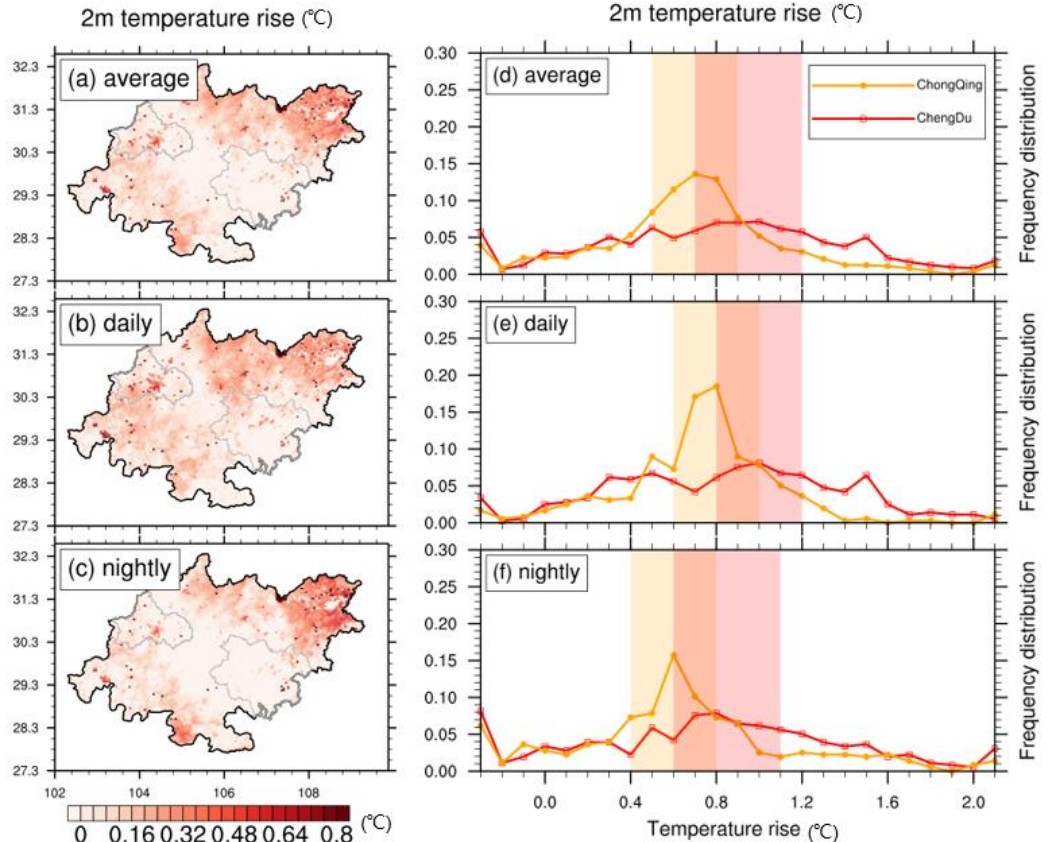

**Fig. 8. The spatial distribution of 2 m-temperature rise due to the urbanization, (a) average; (b) daily; and (c) nightly. And the temperature rises frequency distribution: (d) average; (e) daily; and (f) nightly. (The 'daily' means the average variable during the daytime,'nightly' means the average variable during the nighttime. This description also applies in Fig. 10 below.)**

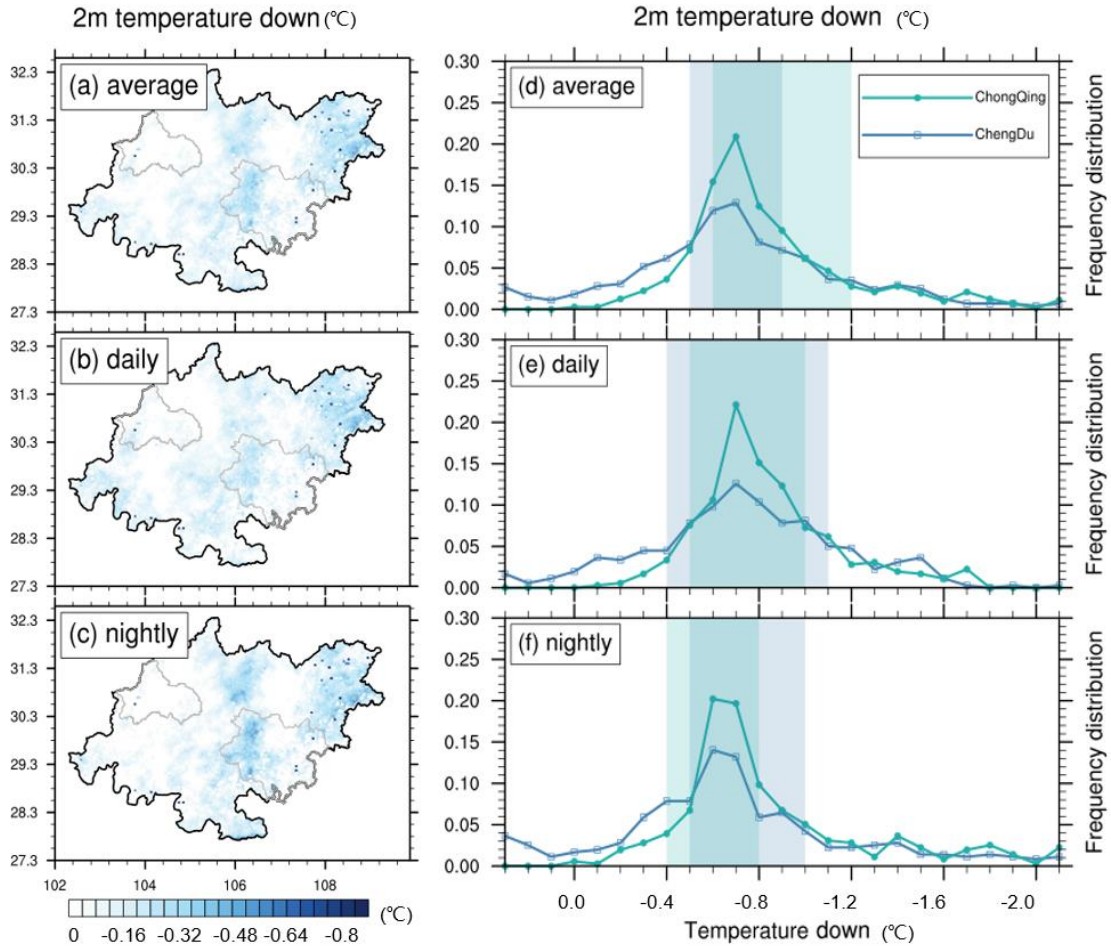

**Fig. 9. The spatial distribution of 2 m-temperature cooling because of the future planning scenario, (a) average; (b) daily; and (c) nightly. And the cooling frequency distribution: (d) average; (e) daily; and (f) nightly.**

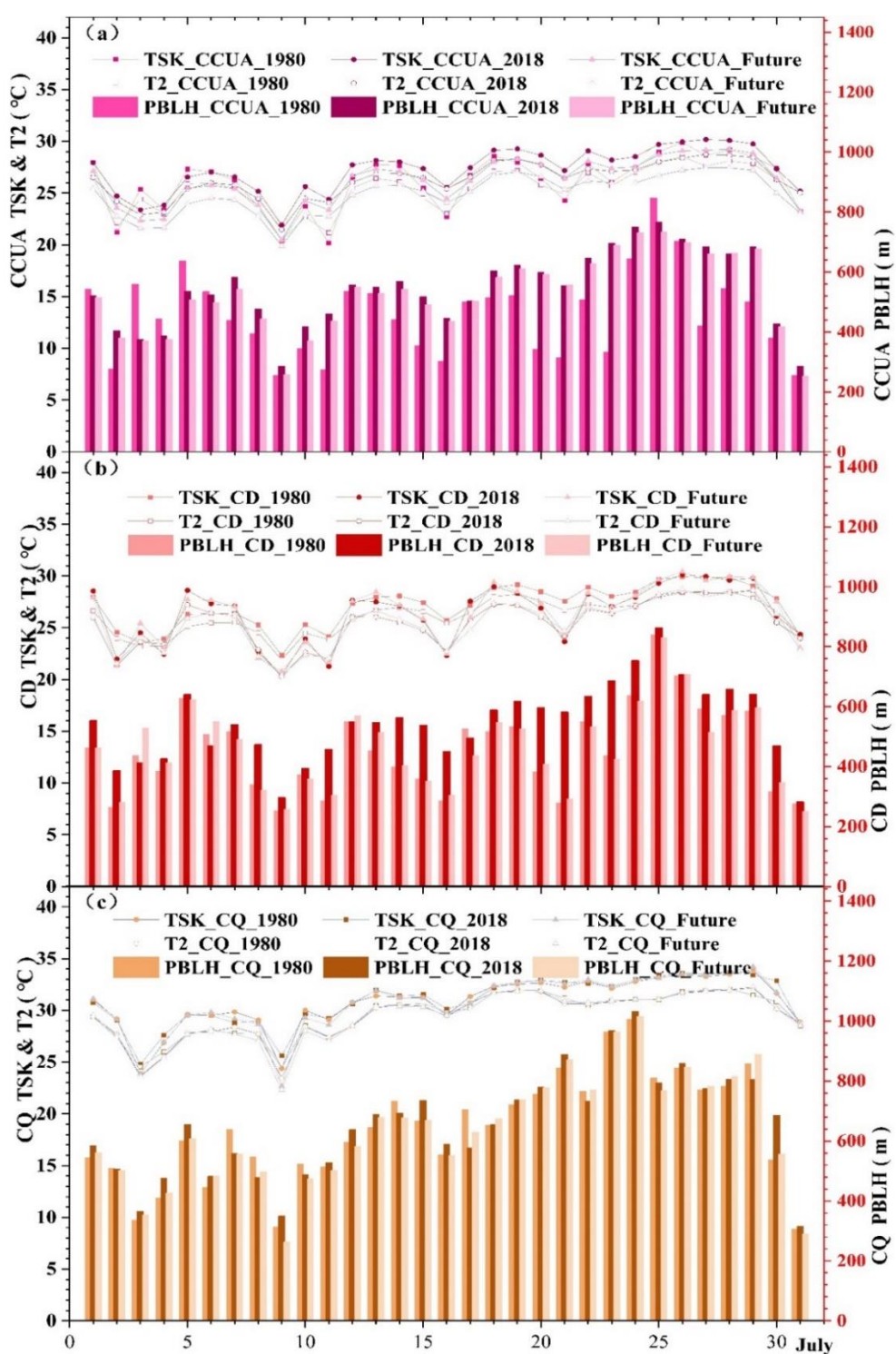

Fig. 10. (a) The TSK and T2 trends, and PBLH distribution over Chengdu-Chongqing Urban Agglomeration; (b)The TSK and T2 trends, and PBLH distribution over ChengDu; (c) The TSK and T2 trends, and PBLH distribution over ChongQing. (The 'trends' means the daily average change of TSK, T and PBLH. The unit of TSK and T is Celsius (℃), the coordinate axis is on the left; The unit of PBLH is meter (m), and the coordinate axis is on the right.)

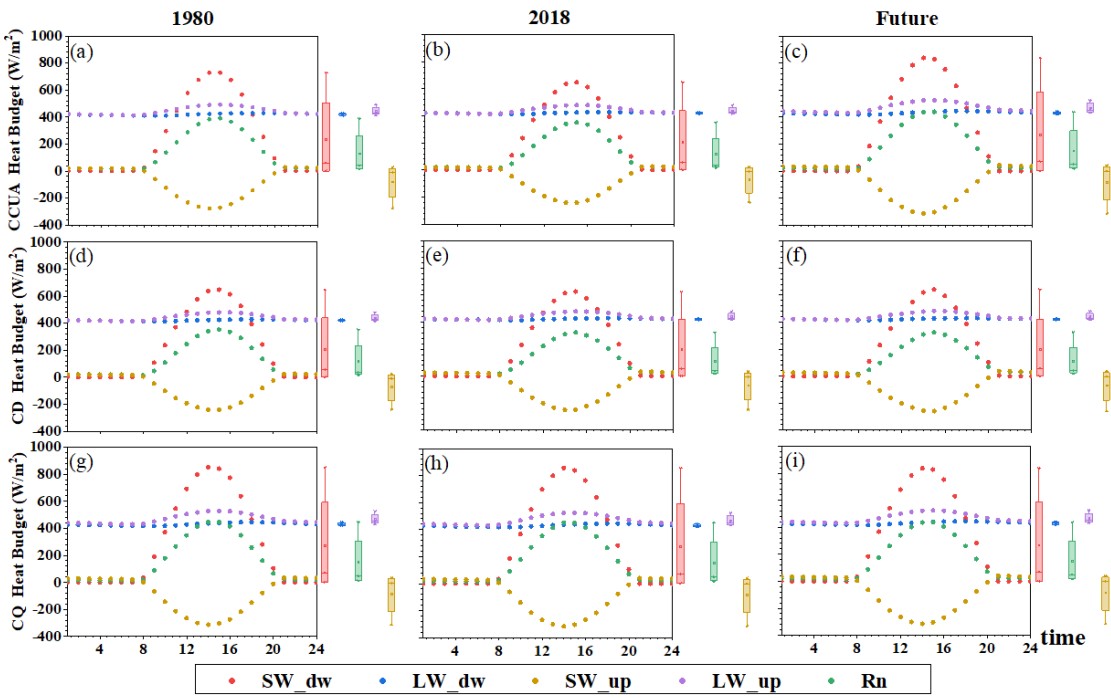

**Fig. 11.** The diurnal variations in the changes in surface energy budget over the land use grids of 1980, 2018, and future runs of Chengdu-Chongqing Urban Agglomeration . (a) 1980, (b) 2018, and (c )Future. The diurnal variations in the changes in surface energy budget over the land use grids of 1980, 2018, and future runs of ChengDu. (d) 1980, (e) 2018, and (f) Future. The diurnal variations in the changes in surface energy budget over the land use grids of 1980, 2018, and future runs of ChongQing. (g) 1980, (h) 2018, and (i) Future.

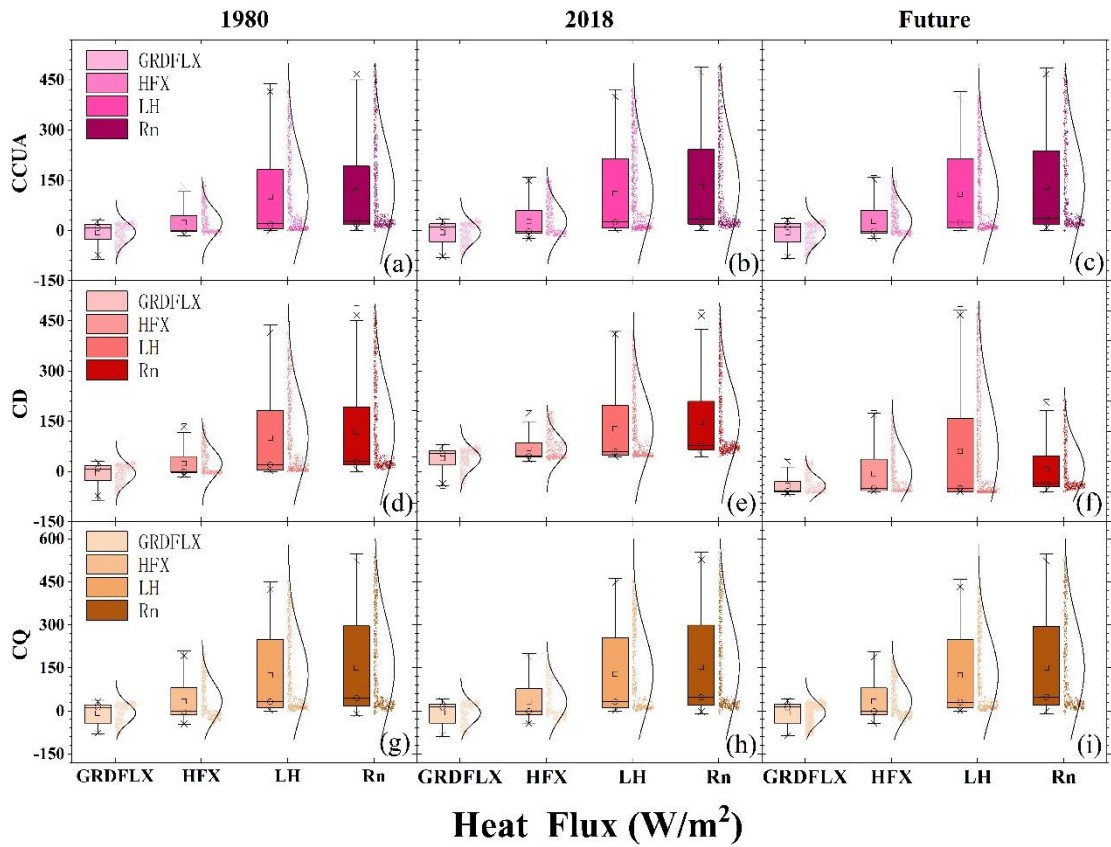

**Fig. 12.** The heat flux distribution changes in surface heat flux over the land use grids of (a) 1980, (b) 2018, and (c) Future of Chengdu-Chongqing Urban Agglomeration; The heat flux distribution changes in surface heat flux over the land use grids of (d) 1980, (e) 2018, and (f) Future of Chengdu; The heat flux distribution changes in surface heat flux over the land use grids of (g) 1980, (h) 2018, and (i) Future of Chongqing.

29