# Peer review of "Impact of urbanization on thermal environment of Chengdu-Chongqing Urban Agglomeration under complex terrain"

_Earth System Dynamics, 2021_

## Author Response (AR1)

**MS No.: esd-2021-22:Impact of urbanization on thermal environment of Chengdu-Chongqing Urban Agglomeration under complex terrain,Si Chen et al.**

**Response to Editor**

Dear Dr. Gabriele Messori,

Thank you very much for your assistances involved in reviewing the manuscript. We revised the manuscript by paying particular attention to the comments as follows:

1. We modified figure 1 by adding sub-figure 1d on the distribution of nearly 2400 meteorological observations station, and the relative description.
2. We add the Fig.3 on the scope of the future land use Scenario designed and planned to illustrate the extension and magnitude of the modifications implemented in the future scenario, and the relative description.
3. We add discussion on future implementation of planning to protect the ecological environment and mitigate the impact of urban expansion on the climate.
4. We add discussion on chooses of parameterization schemes in WRF.

To facilitate this discussion, we first retype the comments in italic font and then present our responses to the comments.

**Comment 1:** *"Following your replies to the Reviewers, I would like to invite you to submit a substantially revised version of your study for further consideration in ESD."*

**Response 1:** "We appreciate your clear and detailed feedback and hope that the explanation has fully addressed all your concerns."

**Comment 2:** *"I would specifically encourage you to contextualise and discuss in the text the implications of any specific modelling choices (e.g. the use or not of the BEP parametrisation), supporting this discussion with references where appropriate."*

**Response 2:** We added the discussion of parameterization scheme as suggested, please see Page2 line35 to Page3 line2 in the revised manuscript.

**Comment 3**: *"Regarding the concerns of Reviewer 2, you should improve the contextualisation of your results by expanding the discussion of how they relate to other urban studies conducted in cities with complex topography, and which parts of your conclusions may have a broader implication than just the case study considered here."*

**Response 3**: The simulations for *Chengdu Chongqing Urban Agglomeration* show that the complex terrain effects the wind system and atmospheric circulation through dynamic process, and then affects the extreme precipitation and thermal environment. This is also true for other cities under complex terrain. We further discussed the effects of the future implementation of planning to protect the ecological environment and mitigate the impact of urban expansion on the climate, to provide help for the

government's planning policies in the future. Please see the details in Section "3.4 Mitigation of future heat stress" from line36 on page 9 to line 27 on page10 in revised manuscript.

**Response to Reviewers#1**

Dear Reviewers#1,

Thank you very much for your time involved in reviewing the manuscript and your very encouraging comments on the merits. To facilitate this discussion, we first retype your comments in *italic font* and then present our responses to the comments.

**Comment 1:** *"This manuscript consists into a relevant study for the study of the urban heat island effect by including the influence of the surrounding topography. The document is well written, presenting a high-quality work in a clear and organized way."*

**Response 1:** We appreciate your clear and detailed feedback and hope that the response has fully addressed all your concerns.

**Comment 2:** *"Before publication, I would like to see a small improvement in this document comprehending the future scenario. As a reader, I felt that there is little detailed information to understand the extension and magnitude of the modifications implemented in the future scenario to induce the results.". "For instance, to a naked eye, such modifications are not visible in figure 2. My suggestion to the authors is to increase the figure 2 (increase also the legend)"*

**Response 2:** We add the Fig.3 on the scope of the future land use Scenario designed and planned to illustrate the extension and magnitude of the modifications implemented in the future scenario. Fig3 is the location diagram of the protection area for future land use planning designed based on "*Chengdu Chongqing Urban Agglomeration Development Plan, 2018*" and "*Guidelines for Ecological Protection and Restoration Project of Mountains, Rivers, Forests, Fields, Lakes and Grasses (for Trial Implementation), 2020*". The two documents include a series of ecological and environmental protection measures such as returning farmland to forests and grasses and expanding lake water system wetlands. Please refer to lines 27-37 on page 4 and lines 1-19 on page 5 in the revised version of the manuscript.

[Figure]

CY_Boundary

Rivers

Land Ecological Corridors

Water System Corridors

Sichuan-Yunnan Forest Reserve

Qinba Biodiversity Ecological Function Zone

Three Gorges Reservoir Water and Soil Conservation Ecological Function Zone

Da-Xiao-Liang Mountain Water and Soil Conservation Ecological Function Zone

Wuling Mountain Ecological Diversity and Soil and Water Conservation Ecological Function Zone

**Fig. 3 Scope of future land use scenarios planned through government policy documents.**

Response to Reviewers#2

Dear Prof. Takebayashi, Hideki

Thank you very much for your time involved in reviewing the manuscript and your very encouraging comments on the merits. To facilitate this discussion, we first retype your comments in *italic font* and then present our responses to the comments.

**Comment 1:** *"In this study, the changes in the thermal environment when the topography and land use were changed were calculated by WRF."*

**Response 1:** We appreciate your clear and detailed feedback and hope that the explanation has fully addressed all your concerns.

**Comment 2:** *"In the model verification in Section 3.1, the cross-correlation of the different variables shown in Fig. 4 is not important, and the bias, RMSE, and correlation coefficient of the observed and calculated values in each valuable should be evaluated. Please present the observation points in Fig. 1, explain the outline of the observation, and discuss the calculation accuracy."*

**Response 2:** We revised as suggested. We add the Fig.1 (d) in Fig. 1 on the distribution of nearly 2400 meteorological observations stations from the China Meteorological Administration. Please refer to Fig.1. Limited by the observation conditions, we only compared the simulated 2m temperature with the observed temperature, and calculated their bias (Fig.3b), RMSE (Fig.3d) and correlation coefficient (Fig.3f).

**Comment 3:** *"It is considered that the wind system has changed due to changes in topography. In particular, consider the effect of mountain and valley wind circulation on the thermal environment. The authors stated that the topography further changes the heat island effect. It must be discussed whether this is a general finding. This appears to be happened a result of setting conditions by the authors. If so, this study is just a case study and it should not be published as an academic research paper."*

**Response 3:** Thank you for your comments. Complex terrain affects the wind system and atmospheric circulation through dynamic process. The mountain and valley wind circulation along urban agglomeration has a great impact on the thermal environment. The Chengdu Chongqing urban agglomeration we studied is in Sichuan Basin and belongs to concave landform. The heat accumulated in the basin under complex terrain during the day is not easy to lose, which will make the heat in the basin higher than that in suburban and mountainous area. The purpose of the experiment is to explore the impact of topographic factors and urbanization factors on the heat island effect, and the impact of the superposition of the two on the urban heat island effect. The results show that the complex terrain further enhance the urban heat island effect located in the expansion area of complex terrain. We further discussed whether the future

implementation of planning to protect the ecological environment and mitigate the impact of urban expansion on the climate will be effective, in order to provide help for the government's planning policies in the future.

We would like to take this opportunity to thank you for all your time involved and this great opportunity for us to improve the manuscript. We hope you will find this revised version satisfactory.

[Figure]

**Fig. 1.   (a)Configuration of the three nested domains for WRF simulation. (b) The original terrain in the simulated area (m), and (c) after smoothing the terrain. (d) Location of meteorological observation stations.**

**Response to Reviewers#3**

Dear Reviewers#3,

Thank you very much for your time involved in reviewing the manuscript and your very encouraging comments on the merits. To facilitate this discussion, we first retype your comments in *italic font* and then present our responses to the comments.

**Comment 1:** *"This is a manuscript on a very important topic of the impacts of urbanization on the microclimate. I have a few major concerns, which the authors may address:"*
**Response 1:** We appreciate your clear and detailed feedback and hope that the response has fully addressed all your concerns.

**Comment 2:** *"1. The write-up is a bit loose and so is the organization. For example, the abstract should clearly state the hypothesis, results, and implications of the results. The modeling framework is not discussed well."*
**Response 2:** Thank you for your sincere suggestions. We will modify the "Abstract" to make the assumptions and conclusions of the summary clearer. We will revise the manuscript as suggested in the revised version.

**Comment 3:** *"2. The details of the WRF simulation zone should be provided with the justification of the selection of such a zone."*
**Response 3:** We revised the manuscript as suggested. The Chengdu-Chongqing Urban Agglomeration (CCUA) is the region with a high potential for economic development in southwest China (Latitude range: 28°10' to 32°25'; Longitude range:101°56′ to 108°57′), surrounded by the Qinghai-Tibet Plateau, Daba Mountain, Huaying Mountain, and Yungui Plateau (Fig. 1a). The surrounding mountains are primarily between 1,000 and 3,000 meters above sea level. However, the urban agglomeration is an extremely sensitive area where a series of ecological and environmental problems are highly concentrated and intensified. the CCUA was rapidly urbanized in the last four decades, has led to a three-fold urban area expansion increased to nearly 9,000 square kilometers, thereby affecting the weather and climate. To investigate the urbanization effects on the thermal environment in the CCUA under the complex terrain and explore the improvement of ideal urban planning policies on urban thermal environmental pressure, we conducted the simulations using the advanced WRF model together with the combining land-use scenarios and terrain conditions. We set up three one-way nested domains in the horizontal direction, with resolutions of 1km, 5km, 25km respectively (Fig. 1a).

**Comment 4:** *"3. The details of WRF parameterization need to be provided with validation and sensitivity analysis."*
**Response 4:** We added the discussion of parameterization scheme, see Page2 line35 to Page3 line2 in the revised manuscript.

**Comment 5:** *"4. I am wondering why authors are not using BEP parameterization for urbanization.*

*Recent studies show that the urban structure does impact the microclimate, for example, they may see https://www.nature.com/articles/s41598-018-22322-9 "*

**Response 5:** We now study the heat island effect caused by the expansion of urban area. It is feasible for us to choose SLUCM urban canopy model as a parametric scheme in previous studies. In the later stage, if we pay more attention to the changes of urban thermal environment caused by building structure details inside the city and urban energy consumption, we will also use BEP model for simulation and comparison verification.

**Comment 6:** *"5. In figure 8, The authors said daily and nightly, does that mean day time and night time, daily may not be the correct term."*

**Response 6:** Thank you very much for your correction. It should be "daytime" and "nighttime". We will make corrections in the revised manuscript.

**Comment 7:** *"6. More detailed validation may be needed. For example, on a heatwave day, how do the hourly variation of simulations and observations match, what are the diurnal variations at different locations etc."*

**Response 7:** Thank you for your suggestion. We used meteorological observations from about 2400 meteorological observation stations (Fig.1d in the revised manuscript) to validate. We interpolated the observation data of these stations to the surface of the whole CCUA and obtained the grid observation data with 1km resolution. We carried out the surface verification of 2m air temperature in the whole area (Fig. 4b, 4d & 4f in the revised manuscript). Finally, we focused on the average daily average hourly variation of 2m air temperature and surface air temperature in the whole CCUA range (broken line diagram in Fig. 10).

**Comment 8:** *"7. Is there any seasonal variation of UHI?"*

**Response 8:** This study only focuses on the impact of urban expansion under complex terrain on the thermal environment of urban agglomeration in summer, since summer is the season with the most obvious urban heat island (UHI) effect of urban agglomeration and the season with the most obvious impact on residential comfort.

We would like to take this opportunity to thank you for all your time involved and this great opportunity for us to improve the manuscript. We hope you will find this revised version satisfactory.

Sincerely,

Si Chen, Zhenghui Xie, et al.

---

## Author Response (AR2)

**Response to Anonymous Referee #4**

Thank you very much for your time involved in reviewing the manuscript and your very encouraging comments on the merits. To facilitate this discussion, we first retype your comments in italic font and then present our responses to the comments.

**Comment 1:** *"On page 5, Lines 25-26: '"No-Topography (Fig. 1c)", that is smoothing the mountainous terrain around the CCUA.' I don't understand what does this (smoothing) means? How this is being processed? From Fig. 1c, it seems the topography surrounding CCUA has been removed (especially the Plateau), am I right?"*
**Response 1**: Yes, the high-altitude mountains around CCUA have been removed, which makes the terrain of the whole simulation area smoother. The purpose is to compare the climate effects with and without complex terrain.

**Comment 2:** "*1. Page 2, lines 27-29: 'The forced initial field data simulated in the model was the re-analyzed data of operational global analysis and forecast data, which are on 0.25×0.25-degree grids, prepared operationally every six hours, from National Centers for Environmental Prediction.' A reference is needed."*
**Response 2**: Revised as suggested.

**Comment 3:** *"2. Page 3, Lines 31-32: ' the land use types of the dataset were reclassified into 17 categories' The default category of USGS for WRF is 24 (or 28 if lakes are considered). How are the 17 categories being processed in WRF?"*
**Response 3**: Due to the slight difference between the LUCC data classification of CAS geographic resource platform (IGSNRR) and USGS classification standard, in order to meet the classification standard in WRF model, we must reclassify the data of IGSNRR according to USGS.

Firstly, we should refer to the naming method and data content format of "index file" and "binary file" under the existing "/ WRF/geog/landuse/" folder to make the binary file required for the WRF model of corresponding new land use data. The specific method is to re classify the "TIFF grid data" of IGSNRR according to the classification standard of USGS, then make the "TIFF data" into "ASCII file", use ArcGIS software for reclassification,; The next job is to convert the "ASCII file" into "binary file" through FORTRAN, and put the binary file into the new corresponding path as followed; After that, we change the path in namelist.wps file to the new path, and finally conduct simulation experiment and analysis.

In addition, because of the CCUA is in subtropical climate area, there are no following land cover types: "mixed dryland/irrigated cropland and pasture, cropland / Woodland mosaic, savanna, deciduous needleleaf forest, evergreen broadleaf forest, evergreen needleleaf forest, herbaceous tundra and wooded tundra", but this will not affect the operation of WRF.

**Comment 4:** "*3. Lines 30-34: Please remove the space after '(' in the cited references.*"
**Response 4**: Revised as suggested.

**Comment 5:** "*4. Fig.1. The value of the color scale of Fig. 1b is missing. Fig. 1d, the quality is too low to show anything.*"
**Response 5**: We had added the value of the color scale of Fig. 1b. We improved the image quality of Fig. 1d and added icons to help understand.

[Figure]

**Comment 6:** "*5. Fig.4. Please check the caption! Please show the difference between the results based on USGS and IGSNRR. Are there any results based on MODIS landcover? It would be interesting to show the difference between results based on MODIS, USGS, and IGSNRR, which could show the importance of using IGSNRR. This could also show hits to readers in their future work.*"
**Response 6:** Thanks for your suggestions. Because this study mainly focuses on exploring the impact of urbanization in complex terrain environment on local geothermal environment, we chose IGSNRR data which is more suitable for local CCUA. In fact, as the data from different sources can show the changes of urban land use types, the results of simulation experiment analysis can support our research conclusions.

For your concern about the difference between the simulation results of land use and landcover data from "USGS" and "MODIS" and even more other land use data with the results from "IGSNRR" data, we will accept your suggestion and conduct a special study in the later stage to explore the impact of different land use data on the heat island effect of urbanization. Thank you very much for your sincere advice. Please look forward to it.

**Comment 7:** "*6. Figs. 8&9, what does 'daily' mean? Do you mean the average during the daytime?*"

**Response 7:** Yes, the 'daily' means the average variable during the daytime.

**Comment 8:** *"7. Fig.9. I would rather use negative values to indicate cooling."*

**Response 8:** Revised as suggested, please refer to Fig. 9.

**Comment 9:** *"8. Fig. 10. What does 'trends' mean? What are the unites?"*

**Response 8:** 'trends' means the daily average change of TSK, T and PBLH. The unit of TSK and T is Celsius (℃), the coordinate axis is on the left; The unit of PBLH is meter (m), and the coordinate axis is on the right.

We would like to take this opportunity to thank you for all your time involved and this great opportunity for us to improve the manuscript. We hope you will find this revised version satisfactory.

**Sincerely,**
**Si Chen, Zhenghui Xie, et al.**

---

## Author Response (AR3)

**MS No.: esd-2021-22:Impact of urbanization on thermal environment of Chengdu-Chongqing Urban Agglomeration under complex terrain,Si Chen et al.**

**Response to Editor**

Dear Dr. Gabriele Messori,

Thank you very much for your assistances involved in reviewing the manuscript. We revised the manuscript by paying particular attention to the comments as follows:

1.  ***Comment 1:*** *While you have provided a complete set of replies to the remaining comments, I note that in many cases the corresponding explanations have not been included in the revised paper text (e.g. comments no. 1 and 3, but there are other examples). Since other readers may have the same questions as the Reviewer, I would encourage you to include at least part of the clarifications you provided to the Reviewer in your updated text.*
    **Response 1:** For 'comments no.1 from Anonymous Referee #4', we added a detailed clarifications in the revised manuscript as suggested. Please refer to 'Page5, Lines 26-30'.
    For 'comments no.3'from Anonymous Referee #4, we added a description on reclassification of land use data in the revised manuscript as suggested, please refer to 'Page3, Lines 31-35' and Table 1 for details.

2.  ***Comment 2:*** *Similarly, while I understand that you may want to avoid significant new analysis following the Reviewer's comment no. 6, this is indeed an interesting point that may be discussed in the concluding section of the text as future work.*
    **Response 2:** We added a discussion in the revised manuscript as suggested, please refer to 'Page11, Lines 20-24'.

3.  ***Comment 3:*** *I also believe that the caption to Fig. 4 is still incorrect (check the panel letters) - a point raised by the Reviewer which you did not update.*
    **Response 3:** Revised as suggested, please refer to Fig.4.

In addition, we checked and corrected the format of the newly added references to meet the requirements; The legend explanation of Fig.8 and Fig.10 is added, and the error identification information of Fig. 9 is corrected. For details, please refer to the revised manuscript.

We would like to take this opportunity to thank you for all your assistances on this manuscript.

**Sincerely,**
**Si Chen, Zhenghui Xie, et al.**